# Ribosome Biogenesis Alterations in Colorectal Cancer

**DOI:** 10.3390/cells9112361

**Published:** 2020-10-27

**Authors:** Sophie Nait Slimane, Virginie Marcel, Tanguy Fenouil, Frédéric Catez, Jean-Christophe Saurin, Philippe Bouvet, Jean-Jacques Diaz, Hichem C. Mertani

**Affiliations:** 1Cancer Initiation and Tumor Cell Identity, Cancer Research Center of Lyon, Université de Lyon, Université Claude Bernard Lyon 1, Inserm U1052, CNRS UMR5286 Centre Léon Bérard, 69008 Lyon, France; sophie.NAITSLIMANE@lyon.unicancer.fr (S.N.S.); virginie.marcel@lyon.unicancer.fr (V.M.); frederic.catez@lyon.unicancer.fr (F.C.); pbouvet@ens-lyon.fr (P.B.); 2Institute of Pathology EST, Hospices Civils de Lyon, Site-Est Groupement Hospitalier- Est, 69677 Bron, France; tanguy.fenouil@chu-lyon.fr; 3Gastroenterology and Genetic Department, Edouard Herriot Hospital, Hospices Civils de Lyon, 69008 Lyon, France; jean-christophe.saurin@chu-lyon.fr

**Keywords:** ribosome, colorectal, rDNA, rRNA, translation, targeting, human, cancer

## Abstract

Many studies have focused on understanding the regulation and functions of aberrant protein synthesis in colorectal cancer (CRC), leaving the ribosome, its main effector, relatively underappreciated in CRC. The production of functional ribosomes is initiated in the nucleolus, requires coordinated ribosomal RNA (rRNA) processing and ribosomal protein (RP) assembly, and is frequently hyperactivated to support the needs in protein synthesis essential to withstand unremitting cancer cell growth. This elevated ribosome production in cancer cells includes a strong alteration of ribosome biogenesis homeostasis that represents one of the hallmarks of cancer cells. None of the ribosome production steps escape this cancer-specific dysregulation. This review summarizes the early and late steps of ribosome biogenesis dysregulations described in CRC cell lines, intestinal organoids, CRC stem cells and mouse models, and their possible clinical implications. We highlight how this cancer-related ribosome biogenesis, both at quantitative and qualitative levels, can lead to the synthesis of ribosomes favoring the translation of mRNAs encoding hyperproliferative and survival factors. We also discuss whether cancer-related ribosome biogenesis is a mere consequence of cancer progression or is a causal factor in CRC, and how altered ribosome biogenesis pathways can represent effective targets to kill CRC cells. The association between exacerbated CRC cell growth and alteration of specific steps of ribosome biogenesis is highlighted as a key driver of tumorigenesis, providing promising perspectives for the implementation of predictive biomarkers and the development of new therapeutic drugs.

## 1. Introduction

In 2018, colorectal cancer (CRC) was ranked the third most frequent cancer world-wide, after lung and prostate cancers in men and lung and breast cancers in women, by the International Agency for Research on Cancer (IARC) [1]. CRC is the most frequent form of gastrointestinal cancer and represents around 10% of all cancers, affecting 1.8 million new individuals each year [1]. The CRC mortality rate is slightly higher in men than in women and results in almost 880,000 deaths per year, making it the second-leading cause of cancer-related deaths [2,3]. CRC is a major health burden, as indicated by the steady increase in the incidence rate of age-standardized CRC cases over the period of 1990–2017 [4] and by more alarming data showing a significant rise in CRC cases in patients under 50 years of age [5]. CRC staging at diagnosis is mostly performed using the TNM system (Tumor–Node–Metastasis) and has a strong prognostic value, with the higher stage being associated with the poorest survival [3]. The 5-year post-operative overall survival rate is 90% in patients diagnosed with localized CRC, 70% in patients with regionalized CRC, and 14% in patients with metastatic CRC [6]. Patients with metastatic CRC have a median overall survival of 30 months [5], often resulting from deleterious metastatic liver and/or lung invasion [7]. Various approaches have been developed to curb the mortality rate of CRC, in particular to improve the early detection using different analyses such as fecal blood tests and colonoscopy, in-depth enquiry into family history, blood analysis for tumor markers [8,9], and analysis of tumor gene expression [10,11]. These practices are also reinforced by a broad panel of imaging methods [12]. Once diagnosed, clinico-pathological characterization of CRC samples is decisive for patient management and treatment choice (chemotherapy and/or irradiation as neoadjuvant or adjuvant treatment), and especially after tumor surgical resection, for assessing follow-up outcome [13,14]. Neoadjuvant chemotherapy with more or less irradiation (for rectal carcinoma) can be performed to downstage CRC before resection [3]. Targeted therapies with relative efficacy are also used to treat CRC tumors with specific molecular alterations [15]. Indeed, the choice of drugs (systemic chemotherapy or targeted treatment) against CRC is largely influenced by the molecular profile determined for each tumor [14]. The molecular characterization of colorectal tumors is based on the analysis of genetic mutations and also on the DNA microsatellite status (i.e., stable/MSS versus unstable/MSI) [16]. Frequent genetic driver mutations occur in CRC cells, enabling the development of specific cytotoxic and/or cytostatic molecules [15,17]. Among the genetic mutations, the *Kirsten rat sarcoma 2 viral oncogene homolog* (*KRAS*) gene mutated in 40% and the *BRAF* gene mutated in 10% of CRC patients are useful predictive markers for metastatic disease [2,18,19]. Metastatic patient treatment including anti-epidermal growth factor receptor (EGFR) monoclonal antibodies is validated for RAS-wild-type tumors (most frequently occurring in the left colon or rectum), whereas vascular endothelial growth factor (VEGF) antibody treatment is proposed for RAS-mutant tumors (most frequently involving the right colon) [20,21,22]. Colorectal carcinogenesis follows a step-by-step process of gene alterations often initiated by inactivating mutations in the tumor suppressor *adenomatous polyposis coli* (*APC*) gene [22]. Somatic APC mutations are present in almost 80% of sporadic human CRC and frequently occur within the coding sequence, resulting in a truncated defective protein [23]. This early-stage mutation is associated with the presence of polyps with low-grade intra-epithelial neoplasia [23,24,25] The consequence of APC inactivation results in aberrant stimulation of the wingless/int1 (WNT) signaling pathway in colonic epithelial cells, leading to a cascade of oncogenic events ultimately forming an invasive colorectal carcinoma [25]. The hyper-stimulation of large components of the WNT signaling pathway was also demonstrated using integrative genomic analyses in large clinical CRC studies [26]. Colonic and rectal tumor progression is further promoted by inactivating mutations in the *TP53* gene, activating mutations of *KRAS* leading to constitutive activation of the epidermal growth factor (EGF) pathway, and inactivating mutations of *SMAD4* resulting in the shut-down of the proliferation inhibitory canonical pathway of transforming growth factor (TGF)-β [27,28,29,30]. More recently, epigenetic alterations including histone modifications, DNA sequence methylation, and expression of non-coding long/micro/circular RNAs has gained attention in CRC studies [31], in particular for understanding the microsatellite instability (MSI)/highly mutated subgroup [10]. However, the strong genetic and phenotypic heterogeneity of CRC samples still represents a challenge for CRC patient stratification, adapting treatment strategies, and managing chemoresistance [32]. Consequently, the identification of reliable diagnostic biomarkers and/or relevant targetable pathways associated with specific CRC subtypes is critically needed.

Among the innovative pathways of interest susceptible of improving CRC patient management, regulation of translation and ribosome biogenesis remain to be revisited based on the recent discoveries in the field. Indeed, extensive data indicate that regulation of translation, and particularly the initiation phase, is of utmost importance for the survival and growth of rapidly dividing cancer cells by providing an adapted quantitative but also qualitative cancer proteome [33,34,35,36]. The process of protein synthesis strictly depends on the elaborate multi-step biogenesis of ribosomes with a precise spatial and functional organization to adjust cell needs [37,38,39]. Our general view on ribosome activity has evolved over the past ten years and the ribosome is no longer considered to be a basic platform for protein synthesis, but also a major regulating switch for gene expression at the translational level in normal [40] and cancer cells [35]. The aim of this review is to present all aspects of ribosome biogenesis alterations reported in human colorectal cancers and explore the possibility of developing neo/adjuvant therapies based on direct or indirect targeting of ribosome production in CRC.

## 2. Ribosome Biogenesis

Human ribosomes are ribonucleoprotein (RNP) complexes comprised of seventy-nine ribosomal proteins (RP) and four ribosomal RNAs (rRNA) [37,38,39,41]. In eukaryotes, ribosome biogenesis is a sequential and highly complex process finely tuned by a spatial and oriented regulation that starts in the nucleus and ends in the cytoplasm [37,38,39,42]. At several stages of cell life, ribosome biogenesis per se could account for more than half of the total energy of the cell [37,38,39]. Therefore, a stringent control of ribosome biogenesis is mandatory to adjust the amount of ribosomes to maintain cell protein synthesis demands according to microenvironmental changes, including nutrient and oxygen availability [38,39].

The mature translating ribosome found in the cytoplasm is organized in two subunits, usually called the large 60S subunit and the small 40S subunit. In humans, the 60S subunit contains the 28S, 5S, and 5.8S rRNAs and 47 RPs, while the 40S subunit contains the 18S rRNA and 33 RPs [37,38,39]. The 28S, 5.8S, and 18S rRNAs are synthesized by the RNA polymerase (RNA pol) I, whereas the 5S is synthesized by the RNA polymerase III [37,38,39]. The 28S, 18S, and 5.8S rRNAs arise from a single pre-rRNA precursor encoded by about 400 rDNA genes localized in the sub-telomeric regions of the acrocentric chromosomes 13, 14, 15, 21, and 22 and organized in tandems. The 5S rRNA is encoded by different clusters of rDNA genes also tandemly organized but exclusively localized on chromosome 1 [37,38,39]. The first steps of ribosome biogenesis take place in particular nuclear domains called the nucleoli. Nucleoli are highly dynamic and transient nuclear domains formed around the nucleolar organizer regions (NOR) of the acrocentric chromosomes when ribosome biogenesis is activated [37,38,39]. Nucleoli are the densest nuclear domains easily visible in phase contrast microscopy. Their density represents the very high local concentration of proteins required for ribosome biogenesis: seventy-nine RPs, four rRNAs, and more than 250 ribosomal assembly factors [37,38,39]. Indeed, not only does ribosome biogenesis require the synthesis of rRNAs and their assembly with some RPs at an early step, but also many co- and post-transcriptional maturation steps of rRNA including a series of cleavages and chemical modifications, such as uridine pseudouridylation, base and ribose methylation. Altogether, these maturation processes ensure the correct tri-dimensional ribosome structure necessary for its main function, translating mRNAs into proteins [37,38,39,43]. As a consequence, alterations of the numerous regulators and actors of the ribosomal machinery, as observed in several pathologies and notably in cancer, dramatically impact the rate and quality of protein synthesis, and ultimately, the production of the relevant proteome from a given transcriptome.

## 3. An Overview of Ribosome Biogenesis in Human Cancers

Numerous studies mainly focusing on the control of rDNA gene expression have highlighted the link between dysregulation of expression of the rDNA genes and cancer. Hyperproliferative cancer cells are by definition cells with perturbed energy homeostasis and increased activity in protein synthesis [34,35,36,44], and thus, an increase in ribosome biogenesis participates in maintaining such a high rate of protein synthesis [45,46,47,48].

### 3.1. Regulation of Ribosome Biogenesis by Oncogenes and Tumor Suppressors

Cancer cells are characterized by constitutive activation of growth signals which alter the activity of major cell cycle regulating transcription factors [49,50], and the increase in ribosome biogenesis in cancer cells has been extensively linked to the sustained activation of RNA polymerases I and III by these cell cycle transcription factors [51,52]. However, in addition to regulating rDNA gene transcription, some oncogenic transcription factors have been shown to coordinate the overall ribosome biogenesis process. For example, the proto-oncogene MYC represents an active hub toward tumorigenesis [50]. Indeed in cancer cells, MYC plays a major role in the production of ribosomes, through the direct activation of rRNA synthesis but also of a gene network which includes genes encoding all rRNA processing factors, RPs, and factors implicated in the translation machinery [53,54,55,56]. The direct binding of MYC to rDNA activates RNA pol I-mediated synthesis of the 47S rRNA precursor [57]. MYC also activates RNA pol II transcription of genes coding for ribosomal assembly factors and for RPs, as well as that of RNA pol III of 5S rRNA [58]. Therefore, the increase in ribosome biogenesis by MYC is not solely restricted to the activation of rDNA transcription but is also mediated by MYC-induced upregulation of RP gene expression. The crucial causal tumorigenic impact of these coordinated transcriptional activations has been elegantly demonstrated in mouse models exhibiting haploinsufficiency for RP genes [59]. As an example, increased expression of RPL14 and RPL28 induced by MYC over-expression in vivo is a major event in lymphoma progression, leading to impaired translation of the cell cycle regulator cDK11, genomic instability, and defective mitosis that directly contribute to lymphomagenesis [59,60]. Interestingly, the combination of chemical inhibitors of ribosome biogenesis and of the mechanistic targeting of the rapamycin (mTOR) signaling pathway, the main driver of lymphomagenesis [61], has proven to be very powerful at inhibiting tumor growth of MYC-driven B-cell lymphoma in vivo in mouse models [62]. Thus, the uncontrolled activity of MYC observed in many types of cancers exacerbates ribosome biogenesis and promotes aberrant translation that sustains tumor progression [63,64]. Additional oncogenes such as mTOR, PI3K, and Akt were also shown to activate rRNA synthesis partly by directly interacting with the formation of the rDNA preinitiation complex (PIC) [63]. Conversely, tumor suppressors including TP53, PTEN, and RB are very potent ribosome biogenesis suppressors [51,52], regulating various levels from rDNA transcription and processing to RPs and accessory factor expression [64].

### 3.2. Regulation of Ribosome Biogenesis by RPs in Cancers

The causal tumorigenic effect of overexpressed *RPL15* and *RPL35* was similarly demonstrated in immune-deficient NOD SCID mouse models of human breast cancer metastasis to the lung and ovary [65]. RPL15 and RPL35 are assembled at an early stage during the processing of the pre-60S subunit and their stoichiometry is crucial for the structure and function of the human ribosome [37,38,39]. Circulating tumor cells (CTCs) from human breast cancer patients that promote metastasis formation were characterized by high *RPL15*/*RPL35* expression associated with increased expression of regulators of translation (i.e., eukaryotic initiation factor 2F targets), *RP* expression, and global protein synthesis [65]. Importantly, this study also showed that in metastatic breast cancer patients expressing sex hormone receptors, characterization of CTCs with high *RPL15/RPL35* expression could discriminate patients with the worse overall survival [65]. This study suggests that increased protein synthesis induced by *RPL15/RPL35* high expression in CTCs contributes to breast cancer progression. However, the impact of high *RPL15/RPL35* expression on ribosome biogenesis remains to be examined, as well as the correlation between inhibition of ribosome biogenesis and a prolonged survival of metastatic breast cancer patients. Similarly, we showed that ribosome biogenesis is increased in a model of MCF-7 mammary cancer cell progression [66] and the work by Prakash et al. [67] demonstrated that genetic inhibition of ribosome biogenesis reduced lung breast cancer metastasis seeding in a model of syngeneic mice [67].

The study of RP expression and activity has gained growing attention in cancer research. Notably, because of the observation that inherited RP gene mutations producing dysfunctional ribosomes are associated with tissue-specific human pathologies named ribosomopathies and with a strong cancer predisposition [68]. Interestingly from the clinical point of view and in the context of translational research, it was demonstrated a long time ago that alterations in ribosome biogenesis occurring in cancer cells could easily be visualized by silver staining of the AgNOR representing several argyrophilic nucleolar proteins that are master regulators of ribosome biogenesis, including, for example, nucleolin (NCL), fibrillarin (FBL), and nucleophosmin (B23) [64,69]. A plethora of studies performed in a large variety of tumor cells from human biopsies unambiguously showed that AgNOR distribution reflected unusual morphology, hypertrophy, and/or an abnormally elevated number of nucleoli that could be a pathological gold standard for cancer cell recognition in routine diagnosis [46,64,70]. However, although it is well-known that nucleoli-derived features (i.e., AgNOR) are a hallmark of cancer cell transformation, these are also present in numerous non-cancerous conditions, justifying why they cannot be used as a diagnostic tool. Albeit, some of these features are used as prognostic tools for some cancers, such as hepatocarcinoma [71] or clear cell and papillary carcinoma, the most frequent kidney cancers [72]. Moreover, a recent report on computer-assisted scoring systems, such as the index of nuclear disruption (iNO score), highlighted that such tools may help to establish a fine description of nucleoli exploitable for cancer diagnosis [73].

## 4. Regulation and Roles of Ribosome Biogenesis in Human CRC

This section presents the current knowledge regarding ribosome biogenesis regulation in intestinal physiology and CRC. The various steps and actors of ribosome biogenesis that are dysregulated in CRC could provide original targets or biomarkers to improve management of CRC patients.

### 4.1. Regulation of Ribosome Biogenesis in Intestinal Stem Cells

The growth and regeneration of the gastrointestinal tract is uninterrupted throughout life and leads in adulthood to the replenishment of 1–10 billion epithelial cells per day and is entirely renewed in ~4–7 days [74]. This constant renewing of gastrointestinal tissue relies on intense stem cell self-renewal, differentiation, and proliferation activities that are sustained by the activation of all the machineries involved in cell growth, division, and protein synthesis, including ribosomes [75]. In mouse organoids, intestinal stem cell differentiation towards enterocytes is accompanied by an increased ribosome biogenesis signature at the transcriptional level [76], and chemical inhibition of ribosome biogenesis using the rDNA transcription inhibitor CX-5461 induces the disruption of the epithelial lining, most likely by stem cell targeting [77]. In addition to contributing to the normal renewing of the gastrointestinal tract, it appears that ribosome biogenesis also promotes colorectal tumorigenesis. In a mouse model of WNT-driven colorectal tumorigenesis, evidence has been provided that defective Notchless-dependent ribosome biogenesis blocked epithelial cell proliferation, imposed cell cycle arrest, and promoted enterocyte differentiation [78]. In humans, evidence of a link between dysregulated ribosome biogenesis and CRC arose from the demonstration that cells from chronic ulcerative diseases associated with a major risk of developing CRC exhibited hypertrophic nucleoli with upregulation of rDNA transcriptional activity [79]. In addition, it was demonstrated that the non-cancerous tissue surrounding colorectal tumor cells constitutes a field of cancerization strongly enriched in a ribosome biogenesis signature, and which is considered to be an early molecular event in human CRC progression [80]. Thus, it appears that dynamic and controlled activation of ribosome biogenesis ensures the homeostatic equilibrium of normal gastrointestinal cells and colorectal cancer cells defined by faulty homeostasis display ribosome biogenesis alterations [47]. These alterations could impact the early stages of ribosome biogenesis (i.e., RNA polymerases activation, rDNA transcription), but also the later steps in ribosome production (rRNA processing, rRNA modifications, rRNA export), extending the possibilities of new ribosome biogenesis-centered targeted therapies in CRC.

### 4.2. Regulation of Early Stages of Ribosome Biogenesis in CRC

Several studies have revealed that ribosome biogenesis could be the integrator and/or final effector of major altered signaling pathways which are often observed in colorectal tumorigenesis. These dysregulations impact every stage of ribosome biogenesis. The tumor suppressors *TP53* and *RB* and the oncogenes *MYC* and *KRAS* have been shown to mainly modulate the early steps of ribosome biogenesis (i.e., rDNA transcription), whereas other factors such as nucleolin, RRS1, pescadillo, or BOP1 mostly alter the late stages of ribosome biogenesis (i.e., pre-rRNA cleavage and ribosomal subunits’ maturation).

A series of studies illustrated that ribosome biogenesis plays a key role in the initial steps of colorectal tumorigenesis. As described above, activation of ribosome biogenesis mostly relies on hyperactivation of RNA polymerase. Frequent overexpression and/or mutations of co-factors or subunits constituting rRNA polymerases have been described in human CRC and are associated with colorectal tumorigenesis by activating ribosome production [81,82,83,84]. The very first step of rDNA transcription induced by RNA pol I in cooperation with UBF and SL1 factors leads to the production of the 47S pre-rRNA and constitutes the rate-limiting step in ribosome biogenesis [85] (Figure 1).

Then, 47S pre-rRNA is cleaved at both ends of the molecule to generate the 45S pre-rRNA which is further cleaved into 5.8S, 18S, and 28S pre-RNAs species [86]. Recently, it was shown that the level of expression of the *45S* pre-rRNA is a CRC prognostic marker significantly associated with poor overall survival in two independent cohorts of eighty primary CRC patients [87]. Moreover, the pharmacological (UBF-interacting Ca^2+^ chelators) or genetic (siRNA) inhibition of 45S pre-rRNA synthesis in nine human CRC cell lines was associated with p53 protein stabilization, cell cycle arrest, and apoptosis, suggesting that inhibition of rRNA synthesis promotes the tumor suppressive response and thus might be an early step in colorectal tumorigenesis [87]. Although the increased expression of the 45S pre-rRNA is synergistic to increased protein synthesis in CRC cells, the rate of synthesis of other rRNA precursors was not determined in this study [87]. Whether the high expression of the 45S pre-RNA is associated with the production of a specific translatome that would drive colorectal tumorigenesis remains an open question.

### 4.3. MYC and RAS Regulations of Ribosome Biogenesis in CRC

CRC cells often display a significant increase in *MYC* expression and/or activity [88], which directly and indirectly stimulates the expression of all components required for protein synthesis, including rDNA, mRNAs, and tRNAs as well as processing and maturation factors contributing to enhanced ribosome production [82,83,84,85,89]. Indeed, in human CRC, most of the essential factors involved in ribosome biogenesis are overexpressed due to high*MYC* expression levels, as exemplified by the *MYC*-target gene, *aryl hydrocarbon receptor,* which is co-upregulated with *MYC* in CRC and promotes HCT-116 CRC cell proliferation by activation of a ribosome biogenesis transcriptional signature [90].

Although it is well documented that ribosome biogenesis can be controlled by growth factor receptors through MAPK/RAS/signaling [52,91] and that constitutive activation of KRAS represents a major determinant of some CRC subtypes, only few studies have explored the contribution of the oncogenic RAS pathway in ribosome biogenesis-mediated colorectal tumorigenesis. Human colorectal cancer cell lines with constitutive KRAS activation display a gene signature enriched in ribosome biogenesis and translation factors [92]. A recent report showed that the cysteine protease calpain-2 is involved in the inhibition of 47S pre-rRNA biogenesis as well as in the disruption of nucleolar integrity in human CRC DLD-1 cells [93]. The inhibitory effect of calpain-2 on 47S pre-rRNA synthesis is abrogated in a cellular model of DLD-1 cells transfected with the constitutively activate KRAS (G13D) mutant [93], indicating that KRAS regulation of ribosome biogenesis could be a major event in CRC. During colorectal tumorigenesis, consecutive to *APC* loss, the tumor suppressor *TP53* gene is frequently lost or mutated, accelerating the onset of tumor formation partly through the loss of inhibition of ribosome biogenesis [45,94]. Activation of the tumor suppressor RB is also a strong contributor to the inhibition of RNA pol I activity at the PIC complex, inducing a rapid reduction in ribosome biogenesis [95].

### 4.4. Nucleolin and Regulatory Protein Homolog RRS1 in CRC

Nucleolin (NCL) is a multifunctional RNA-binding protein which activates rRNA transcription and pre-rRNA processing [96]. Recently, a novel interaction partner of NCL was discovered, the long non-coding RNA cytoskeleton regulator RNA (CYTOR), which is functionally involved in human CRC tumorigenesis [97]. The CYTOR–NCL interaction activates the nuclear factor (NF)-κB pathway in CRC, and CYTOR and NCL, both overexpressed in human CRC, are associated with poor patient prognosis [97]. It would be of great interest to examine ribosome biogenesis status in these tumors. Additional experiments are now required to validate NCL as a potential target of ribosome biogenesis in CRC.

In eukaryotes, the ribosome biogenesis regulatory protein homolog (RRS1) cooperates with RNA pol III and the protein ribosome production factor 2 homolog (RPF2) to drive 5S rRNA synthesis and also participates in the maturation steps of the pre-60S particle [98]. The biological role of RRS1 was shown by experimental depletion of *RRS1* in mouse embryonic fibroblasts, resulting in impaired ribosome biogenesis-associated nucleolar stress (see Section 4.6) and senescence [99]. The expression of *RRS1* is significantly upregulated in a clinical CRC cohort, associated with tumor aggressiveness and poor overall survival [100]. *RRS1* expression is correspondingly overexpressed in The Cancer Genome Atlas (http://cancergenome.nih.gov) database. The silencing of the *RRS1* gene in two CRC cell lines, HCT-116 and RKO, inhibited cell proliferation, and induced cell cycle arrest and apoptosis. These effects were also demonstrated in vivo on subcutaneously implanted colorectal tumor cells, indicating a key role for the ribosome biogenesis factor RRS1 in promoting tumorigenic properties [100]. Although the consequences of *RRS1* overexpression on the regulation of ribosome biogenesis per se was not determined, other reports showing that high *RRS1* is oncogenic in breast [101,102], thyroid [103], and liver [104] cancers, clearly indicate that RRS1 represents a new promising target to be considered in CRC therapy.

### 4.5. Late Stages of Ribosome Biogenesis in CRC

As stated above, pre-rRNA processing is characterized by an elaborate succession of cleavage, folding, protein associations, and chemical modifications of rRNAs [39,105]. One important step of pre-rRNA cleavage involves the formation of the trimeric complex PeBoW composed of three proteins, the Pescadillo homolog 1 (PES1), the block of proliferation (BOP1), and the WD-repeat domain 12 protein (WDR12) [106]. In mammalian cells, the PeBoW complex is essential for ITS-2 processing of the 32S pre-rRNA into 28S and 5.8S rRNAs, and is mandatory for the formation and assembly of the 60S subunit [106,107,108,109,110]. Additionally, two RNA helicases from the DEAD-box helicase family, DDX21 and DDX27, associate with the PeBoW complex and participate in structural rearrangements that accompany the formation of the pre-60S subunit [111]. The activity and distribution of the PeBoW complex is synchronized with cell cycle progression, and perturbations in its organization were reported to be key mediators of nucleolar stress, cell cycle arrest, and p53-dependent apoptosis [109,110].

#### 4.5.1. PES1 and Interacting Partners in CRC

*PES1* is a MYC target gene [112] that is significantly increased in clinical samples of human CRC tumor cells vs. normal adjacent epithelial cells or normal colon [113]. The suppression of *PES1* in various human CRC cell lines significantly reduced their proliferation and colony-forming ability on soft agar, as well as their growth in vivo following xenotransplantation in immune-deficient mice [113]. Xie et al. also showed that increased *PES1* expression in CRC cells was associated with resistance to chemotherapeutic treatments (i.e., etoposide, 5-FU, doxorubicin, vincristine) and provided protection against DNA-induced damage [114]. The regulation of *PES1* transcription in HCT-116 cells is mediated via the activation of the c-Jun NH2-terminal kinase (JNK) signaling pathway, indicating that inhibition of colorectal tumorigenesis by JNK inhibitors could also be mediated by a downregulation of *PES1* leading to a decrease in ribosome biogenesis [113].

PES1 and DDX21 form with the G protein nucleolar 3 (GNL3), a complex involved in late processing steps of the 32S pre-rRNA to 28S rRNA, before the incorporation of the latter into the 60S particle [115]. GNL3 is overexpressed in CRC tumor cells compared to normal colon tissue and is significantly associated with poor patient overall survival [116]. The overexpression of GNL3 in HT29 CRC cells activates the WNT signaling pathway, cell proliferation, colony formation, epithelial-mesenchymal transition (EMT), migration, invasion, and in vivo tumor growth, whereas its suppression by siRNA reverses these effects [116]. This study indicates that the contribution of GNL3 to colorectal tumorigenesis could be mediated by altered ribosome production [116] and further analyses of ribosome biogenesis and protein synthesis rates under high *GNL3* expression is now required.

The long non-coding RNA, circular antisense non-coding RNA in the INK4 locus (*circANRIL*) is a newly identified negative regulator of PES1 activity, which binds to the C-terminal domain of PES1 and inhibits its action on the cleavage of the 32S pre-rRNA [117]. *circANRIL* appears to play a broader role in ribosome biogenesis since it is also an interacting partner of the nucleolar protein NOP14 which is essential for the formation of the 40S subunit [117]. In human embryonic kidney HEK-293 cells, the overexpression of *circANRIL* impairs ribosome biogenesis, induces an accumulation of premature 32S and 36S rRNAs, inhibits cell proliferation, and increases cell death [117]. Interestingly, using primary cultures of human smooth muscle cells and macrophages as a model of atherogenesis, the authors demonstrated that impaired ribosome biogenesis and nucleolar stress-induction due to high levels of *circANRIL*, confers a protective response against atherosclerosis [117]. This mechanism of regulation of ribosome biogenesis through circular lncRNA provides exciting opportunities for investigating ribosome biogenesis in a different yet complementary context to its implication in the regulation of translation in CRC [118].

In addition to the role of PES1 and its partner in regulating ribosome biogenesis, an extra-ribosomal role for PES1 was recently described as a direct activator of human telomerase reverse transcriptase (hTERT), resulting in telomere length maintenance and senescence in breast and liver cancer cells [119]. It would be interesting to examine the possible association between the high hTERT activity in CRC [120] and *PES1*. Hence, the central position of PES1 within a protein-rRNA network integrating ribosome biogenesis, telomerase activity, proliferation, and apoptosis, makes it a target of choice for the future developments of targeted CRC therapies.

#### 4.5.2. Contribution of BOP1 to CRC Tumorigenesis

The PeBoW component BOP1 was also shown to mediate activation of ribosome biogenesis and particularly during the processing of the 47S pre-rRNA to mature 18S and 28S rRNA activated by MYC [121]. It has been reported that the number of copies of the *BOP1* gene present on the 8q24 chromosomal region is amplified in ~40% of human primary CRC and associated with consecutive overexpression of *BOP1* mRNA [122]. Interestingly, the *BOP1* gene is close to the *MYC* oncogene, but its overexpression is more frequent and independent of *MYC* amplification, suggesting that BOP1 overexpression may be a major cause of 8q24 chromosomal region amplification in human colorectal tumorigenesis [122]. Another study confirmed the gain in the 8q24 chromosomal region of the *BOP1* gene and its increased protein expression in frozen micro-dissected human rectal cancer cells [123]. In addition, liver metastases formed in immune-deficient SCID/NOD mice after spleen injection of various CRC cell lines with constitutively activated Wnt/β-catenin pathway, show a specific enrichment in *BOP1* expression [124]. In human CRC clinical samples, *BOP1* is overexpressed in cancer cells compared to normal adjacent epithelial cells and represents a biological marker associated with tumor progression and formation of distant metastases [125]. Very interestingly, the experimental ablation of *BOP1* in the human DLD-1 CRC cell line resulted in altered chromosomal segregation and aberrant mitosis that induce chromosomal instability (CIN) distinctive of CRC cells [126], indicating the importance of BOP1 in maintaining intestinal cell physiology. The overexpression of *BOP1* in SW480 CRC cells stimulated extracellular matrix invasion and two-dimensional (2D)-migratory properties and was accompanied by the activation of the EMT program [124]. In immune-deficient SCID/NOD mice, the splenic injection of SW620 CRC cells lacking *BOP1* resulted in a significant decrease in the number of liver metastases [124]. Similarly, the overexpression of *BOP1* in the HCT-116 CRC cell line stimulated their migratory and invasive properties concomitant to the activation of matrix metalloproteases MMP2 and MMP9, whereas *BOP1* ablation in HT29 blocked their migratory and invasive capacities [125]. Moreover, the lncRNA colon cancer-associated transcript 2 gene (*CCAT2*) was recently related with *BOP1* activation, chemoresistance to 5-fluorouracil, oxaliplatin, and with colorectal tumorigenesis [127]. Microarray and mass spectrometric analyses determined that ribosome biogenesis and translation factors were upregulated in *CCAT2*-overexpressing CRC cells, however the precise status of rRNA synthesis remains to be characterized [127]. The mechanism through which BOP1 activates colorectal tumorigenesis is mediated by the increased expression of active *aurora kinase B* [127], and phenotypic changes associated with migration and invasion are mediated by the activation of the JNK signaling pathway [124,125].

The contribution of BOP1 to colorectal tumorigenesis could be further explored in CRC cells that have lost p53 activity, since the inactivation of *BOP1* in mouse TP53-KO 3T3 fibroblasts is associated with increased sensitivity to camptothecin cytotoxic treatment [128]. This highlights the potential usefulness of combining molecules derived from camptothecin like irinotecan frequently used to treat CRC with drugs that would target the PeBoW complex and/or ribosome biogenesis.

#### 4.5.3. WD12: The Third PeBoW Constituent Deserving Further Attention in CRC

Likewise, WD repeat domain 12, the third stable constituent of the PeBoW complex that participates in 32S rRNA processing was shown to drive tumor progression in glioblastoma cell lines and is a clinical marker associated with poor prognosis in glioblastoma patients [129]. Sun and Qian screened the NCBI GEO database (https://www.ncbi.nlm.nih.gov/geo/) and identified *WDR12* as a key factor upregulated in CRC cells vs. normal adjacent tissue [130]. Therefore, the role of WDR12-mediated alterations in ribosome biogenesis in human colorectal tumorigenesis warrants further investigations.

### 4.6. Regulation of Ribosome Biogenesis by Ribosomal Proteins in CRC

The formation of mature ribosomes is dependent on the orderly assembly of seventy-nine RPs [37,38]. In addition to their function in rRNA processing, some RPs also play extra-ribosomal roles during development, immune responses, and tumorigenesis [131]. The quantitative variations in RP expression levels described in CRC were mainly associated with extra-ribosomal effects leading to what is recognized as the nucleolar or ribosomal stress [132]. The nucleolar stress is caused by genetic mutations or altered expression levels of RP which are consequently not incorporated in ribosomes but bind to the E3 ubiquitin ligase, MDM2, thereby inducing stabilization of p53 and apoptosis activation [52,133]. Diseases associated with RP gene mutations are known as ribosomopathies and are often linked to cancer predisposition [46,68,94,131,134,135,136,137]. Here, we examine the expression and rRNA processing activity of some RPs and how it could be related to experimental and clinical CRC.

Indeed, some RPs are not only implicated in ribosome structural building but are also critical in rRNA processing [37,38]. For instance, RPS20 is involved in maturation of the small pre-40S particle, in cytoplasmic export of the 20S rRNA precursor and in 18S rRNA processing [135,138]. Genetic analysis and exome sequencing indicated that *RPS20* expression is dependent on an inactivating germline mutation that strongly predisposes humans to some forms of nonpolyposis CRC [139]. The authors demonstrated that experimental inactivation of *RPS20* in Hela cells recapitulated late pre-rRNA processing and 18S rRNA maturation defects initially characterized in nonpolyposis CRC clinical samples, providing a causal link between disturbed ribosome biogenesis and CRC predisposition [139].

RPL14 controls the processing of the 45S pre-rRNA and 12S rRNA leading to the production of the mature 5.8S rRNA, and RPS17 controls the late processing stage of the 21S rRNA to nuclear 18S pre-rRNA [140]. Interestingly, RPL14 and RPS17 both potentially contribute to colorectal tumorigenesis [141]. Indeed, *RPL14* and *RPS17* are two of the few genes activated in CRC and associated with microsatellite instability (MSI) markers and inactivation of mismatch repair genes [141]. However, whether MSI in human CRC is dependent on a dysregulation of ribosome biogenesis through upregulation of *RPL14* and *RPS17* is unknown.

Additional RPs, RPS6 and RPS7, both involved in 5‘ETS cleavage to generate the 30S rRNA [140], contribute to human colorectal tumorigenesis in vitro [25,26]. RPS6 expression is upregulated in CRC [132], stimulates cell proliferation, colony formation, and mediates resistance to the MEK1/2 kinase inhibitor selumetinib of a large panel of human CRC cells [142]. RPS7 is also overexpressed in human CRC cells and mainly appears to exert extraribosomal activation of genes linked to tumor hypoxia and glycolysis [141]. The impact of overexpressed RPS6 and RPS7 on CRC ribosome biogenesis has so far not been examined and warrants further investigation.

RPS24, which is necessary for the formation of the 18S rRNA [140], is also an *RP* gene significantly overexpressed in human CRC [143]. RPS24 has been shown to stimulate proliferative and migratory capacities of CRC cell lines HT-29 and HCT-116 [144], and determining the status of ribosome biogenesis will be pivotal in understanding the role of RPS24 in colorectal tumorigenesis.

RPL15 acts as a ribosomal assembly factor essential for the formation of the 60S subunit [145] and is also directly involved in pre-rRNA processing at the internal transcribed spacer 1 (ITS1) site of the 47S pre-rRNA [146,147]. *RPL15* is significantly upregulated in LoVo, HCT-116, SW-480, and SW-620 CRC cell lines compared to non-transformed epithelial cells, and screening of the ONCOMINE 3.0 database (www.oncomine.org) indicated that *RPL15* is overexpressed in CRC and associated with disease progression [148]. *RPL15* inhibition by siRNAs induced a striking reduction of the pre-60S subunit and is associated with cell cycle arrest at the G1-G1/S phase and apoptosis in HCT-116 CRC cells [148]. It would be interesting to define the translational regulation associated with increased synthesis of RPL15 and 60S subunit in colorectal tumorigenesis in order to find novel CRC targets. Similarly to CRC, high levels of *RPL15* expression were found in human gastric cancer and shown to be involved in gastric tumor progression [149]. These data indicate that some RPs represent new potential targets to counteract the hyperactivation of ribosome biogenesis in CRC. The implication of PeBoW complex, RPs, and ribosome biogenesis processing factors upregulated in CRC is summarized in Figure 2.

### 4.7. Ribosome Biogenesis Processing Factors in CRC

Several studies have shown that human CRC progression is associated with the dysregulation of proteins besides RPs but also involved in ribosomal processing. These factors play an important role in the correct processing of rRNAs and ribosome assembly alongside RPs to achieve the production of a functional ribosome [156]. For example, the NIN1 (RPN12) binding protein 1 homolog, also known as NOB1, is an endonuclease which cleaves the 3′ end of the 18S rRNA and controls the final maturation step of 18S rRNA [39]. Additionally, the cleavage of the 3′ end of the 18S rRNA by NOB1 is potentiated by the specific binding of the ribosomal biogenesis factor named “partner of NOB1” or PNO1, which induces a conformational change of NOB1 that increases its binding affinity and activity on the 18S rRNA (Figure 2) [157]. The expression of PNO1 was recently investigated in human CRC by microarray assays, RT-qPCR, and tissue microarray (TMA), and was shown to be overexpressed in cancer cells vs. adjacent normal tissue and associated with poor patient prognosis [151]. The overexpression of *PNO1* in HT-29 and HCT-8 CRC cells prevented apoptosis and stimulated their proliferation in vitro. Moreover, the knock-down of *PNO1* in HCT-116 and RKO cells significantly reduced tumor growth in vivo [151]. In this study, the link between the oncogenic effects of PNO1 and disturbed ribosome biogenesis was subsequently demonstrated by polysome profiling of rRNAs from *PNO1*-ablated HCT-116 cells, and indicated a significant decrease in the amount of 18S rRNA, 40S subunit, 60S subunit, and mature 80S ribosome [151]. The ablation of *PNO1* also resulted in the reduction of global protein synthesis and restored p53 functionality [151]. Further exciting work should now clarify the translational mRNA targets regulated by high levels of PNO1/NOB1 expression in CRC cells and determine their clinical relevance.

Human U3 protein (UTP) 14a (hUTP14a) is a nucleolar protein associated with U3 snoRNA and the DEAH-box RNA helicase DHX37, and that is required for 18S rRNA processing and 40S subunit synthesis [158]. hUTP14a has been shown to participate in the formation of a nucleolar complex that inhibits MYC degradation, thereby promoting its activity during colorectal tumorigenesis [152]. In parallel, nucleolar hUTP14a binds to p53 and RB and stimulates their degradation [159]. It was reported that the nucleolar hUTP14a is significantly overexpressed in CRC TMA sections compared to adjacent normal epithelia, and the co-overexpression of both MYC and hUTP14a is a marker of poor prognosis in CRC [152]. The formation of a stable complex between hUTP14a and MYC supports the proliferation of HCT-116 CRC cells, whereas suppression of *hUTP14a* inhibits HCT-116 cell proliferation in vitro and in vivo after skin implantation in immune-deficient NOD/SCID mice [152]. Further analysis of high *hUTP14a* expression on ribosome biogenesis and general translation regulations should determine whether hUTP14a is a potentially meaningful target in human CRC.

The Shwachman-Bodian Diamond syndrome (SBDS) protein plays a dynamic structural and functional role in the late processing of the large 60S ribosomal precursor by association with the 28S rRNA, and is involved in the production of mature 80S ribosome [160]. Using CRC TMA sections, it has been shown that SBDS is overexpressed in tumor cells compared to normal adjacent cells and high SBDS expression is associated with an unfavorable prognosis [161]. The suppression of *SBDS* induced a significant p53-mediated decrease in HCT-116 cell growth and invasion [161] and further investigations may shed light on the link between SBDS expression and ribosome biogenesis dysregulation in colorectal tumorigenesis. Ribosome biogenesis processing factors that are upregulated and that could potentially be targeted in CRC are indicated in Figure 2.

### 4.8. Chemical Modifications of rRNA in CRC

rRNAs constitute the translational platform on which the mRNA decoding and peptidyl transferase activities are physically connected and functionally controlled [43]. Several types of base or nucleotide modifications accompany various late steps of eukaryotic rRNA biosynthesis and stabilize the three-dimensional (3D) structure of functional ribosomes [43]. However, evidence that altered chemical modifications affect a targeted set of translated mRNAs during development and disease has provided further insight into the impact of qualitative rRNA modifications on ribosome function [40].

The addition of a methyl group to the 2′-hydroxyl group of a ribose (2′-O-ribose methylation) catalyzed by fibrillarin (FBL) on 106 possible sites and the isomerization of uridine to pseudouridine (Ψ) catalyzed by dyskerin (DKC1) on 97 possible sites, are the most frequent 18S, 5.8S, and 28S rRNA chemical modifications [43]. Interestingly, the level of ribose methylation and uridine pseudouridylation at individual sites within the decoding and peptidyl transferase centers has been linked to major mRNA-specific translational defects that could drive tumorigenesis [40,43,162]. We have previously shown that the level of 2′-O-ribose-methylation at some given sites is sensitive to variation in *FBL* expression and influences the preferential translation of oncogenic Internal Ribosome Entry Sites (IRES)-containing mRNAs, like the IGF-IR in MCF-7 breast and in colorectal HCT-116 cancer cells [163]. The level of 2′-O-ribose methylation on human rRNAs in HCT-116 cells with suppressed *FBL* has been established and has resulted in the identification of sites of 2′-O-ribose-methylation vulnerability on the 18S, 5.8S, and 28S rRNAs, that are strongly dependent on FBL activity [164]. This work provides a great opportunity to further study the impact of rRNA 2′-O-ribose-methylation in experimental colorectal tumorigenesis and in human clinical CRC samples.

The C/D-Box small nucleolar RNAs (SNORD) are a conserved family of non-coding snoRNAs which guide, for example, the enzyme FBL to specific 2′-O-ribose methylation sites on rRNAs [165,166]. In human Hela cells, SNORD16 is the snoRNA which directly interacts with the 18S rRNA and guides FBL to methylate the 2′O-ribose on the 18S-Am484 site [167]. It was recently shown that SNORD16 is a molecular marker of human CRC and a driver of colorectal tumorigenesis [154]. SNORD16 overexpression is significantly correlated with age, cancer cell invasion, patient history of colon polyps, and is associated with a poor patient overall survival [154]. HCT-116 and SW-620 cells transduced with lentiviral-*SNORD16* exhibited significant increases in cell proliferation, colony formation, and migratory and invasive capacities [154]. Although not determined in this study, the impact of SNORD16 on colorectal tumorigenesis is likely to be mediated through altered rRNA 2′-O-ribose methylation profiles and a translational reprogramming which could provide new targets when developing CRC therapies.

Base methylations represent another type of rRNA chemical modification which occurs in late stages of ribosome biogenesis and which are mostly involved in maintaining the ribosome translational fidelity [43]. The human nucleolar enzyme NSUN5 is the methyl transferase involved in cytosine methylation on C^5^ position of residue C_3782_ of the 28S rRNA (18S-m_5_C3782) and this chemical modification is necessary for stabilizing the peptidyl (P) transferase site [168]. Interestingly, the expression of *NSUN5* is upregulated in human CRC and associated with disease progression [153]. Experiments using HT29 and RKO CRC cell lines demonstrated that *NSUN5* expression promoted cell proliferation by controlling the expression and activity of major cell cycle regulators in vitro as well as in vivo [153]. The alteration of the level of 18S-m_5_C3782 in CRC samples was however not determined, nor was the translational profile mediated by overexpressed *NSUN5* [153]. Future experiments with CRC cells overexpressing *NSUN5* may provide important insights into the implication of rRNA base modifications in colorectal tumorigenesis.

Crucial evidence of the importance of rRNA modifications in CRC was recently unveiled by the discovery of the reduced frequency of a single nucleotide variation in the 18S rRNA, present in 46% of CRC samples of four independent large cohorts (~10,000 patients) compared to patient-matched normal epithelium (n = 708) [155]. The nucleotide alteration was found by screening changes in the average variant allele frequency on rRNAs and occurs on uridine U1248 of the 18S rRNA which can display a chemical modification, a 1-methyl-3α-amino-α-carboxyl-propyl pseudouridine (m^1^acp^3^Ψ), thus leading to a decrease in 18S-m^1^acp^3^ΨU1248 level in CRC patients [155]. The ribosome biogenesis protein Tsr3 is an enzyme involved in this nucleotide modification which occurs in the peptidyl transferase center [169]. Suppression of Tsr3 in HCT-166 cells reduced the level of m^1^acp^3^Ψ to the one found in CRC samples [155]. The reduced level of m^1^acp^3^Ψ is predicted to alter the structure of the P site and resulted in an enrichment in a proliferative and translational gene signature at transcriptional or translational levels, that could drive colorectal tumorigenesis [155]. All these data show that subtle rRNA modifications originally thought to be structural elements of the ribosome can generate specific ribosomes with preferential translation of mRNAs coding for proliferative factors. Therefore, the targeting of enzymes which catalyze these rRNA chemical modifications may represent a valuable therapeutic tool. The chemical modifications and associated enzymes that are distinctive of CRC are indicated in Figure 2.

## 5. Targeting Ribosome RNA Synthesis in Colorectal Cancer

The rationale for targeting ribosome biogenesis in cancer is based on experimental and clinical evidence showing that tumorigenesis is associated with quantitative increases in ribosome production and/or production of qualitatively-altered ribosome species [44,45,47,48,52,170,171,172,173]. These characteristics have formed the basis for the development of new drugs that disrupt the activation of rRNA synthesis by directly targeting the formation and activity of the RNA pol I transcriptional complex on rDNA [51,174,175,176]. Effective ribosome biogenesis inhibition in cancer cells has gained considerable attention with the development of the two inhibitors, CX-5461 and CX-3543, which selectively bind to rDNA-enriched G-quadruplex regions and halt ribosome production [51,175]. These new drugs that target rRNA transcription and induce cell cycle arrest could represent a novel approach in the treatment of human CRC. Compared to existing chemotherapy using oxaliplatin, 5-fluorouracil, and camptothecin that target rRNA production in an unselective manner, or actinomycin D that binds GC-rich regions of rDNA [177,178], CX-5461 and CX-3543 appear to kill in particular cancer cells or cancer stem cells that have high activation status of ribosome biogenesis. It is also important to note that, in contrast to most other chemotherapeutic molecules used in CRC treatment, oxaliplatin efficacy depends more on the activation of nucleolar ribosome stress than on the induction of a DNA damage response (DDR), further indicating the relevance of targeting ribosome biogenesis in colorectal tumorigenesis [178].

In the p53-wild-type HCT-116 CRC cell line, CX-5461 was shown to inhibit RNA pol I transcriptional activity by disrupting the binding of SL1/TIF-IB transcription factors to the rDNA promoter on the PIC complex, without affecting general transcription and global protein synthesis, thereby promoting nucleolar stress, stabilization of p53, and cell death [179,180]. Interestingly, the cytotoxic effect of CX-5461 in HCT-116 CRC cells is potentiated by cellular DDR induced by ionizing radiation treatment [181]. The recent demonstration in HCT-116 cells, that CX-5461 induces a prominent intracellular DNA damage through inhibition of topoisomerase II activity, indicates that DDR could be the major mechanism of CRC cell death induction [182]. Similarly, in p53 mutant HT-29 and COLO-205 CRC cell lines, CX-5461 treatment is inducing apoptosis [179], but possibly through a mechanism that triggers replication stress and DDR activation, as reported in high-grade serous ovarian cancer [183]. CX-5461 is in phase I/II clinical trials for hematological cancers [184], but further investigations are necessary to understand the crosstalk between ribosome biogenesis inhibition and DDR activation induced by CX-5461 treatment of CRC cells.

The other small molecule which inhibits RNA pol I activity, CX-3543, or quarfloxin, is a fluoroquinolone derivative which interferes with the binding of nucleolin on rDNA G-quadruplex regions, thereby inhibiting RNA pol I-driven transcription [185]. CX-3543 induced in vitro apoptosis of various human CRC cell lines, including p53 mutant COLO-205, HCC-2998, HCT-15, and KM12 cells, and p53-wild-type HCT-116 cells, and inhibited in vivo HCT-116 tumor xenograft growth [185,186]. CX-3543 has entered a phase II clinical trial in patients with low to intermediate grade neuroendocrine tumors [186,187]. Besides, the demonstration that it also causes a strong inhibition of *MYC* expression in CRC cells [186] should stimulate more comprehensive work to assess its impact on colorectal tumorigenesis. It has also been shown that induction of HCT-116 and DLD1 colorectal cancer cell death induced by CX-5461 and CX-3543 treatment in vitro and in tumor xenografts is mediated by the activation of a robust DNA damage response [181]. This type of genotoxic cell response indicates that CX-5461 and CX-3543 could be very potent in killing CRC cells with a defective homologous recombination pathway. This default is generally due to mutations in DNA damage repair enzymes or in *BRCA1/2* genes and is frequently observed in the sub-group of CRC with MSI [10]. Patients with high MSI CRC show a positive response to immunotherapeutic treatment [188], and it will be meaningful to investigate whether CX-5461 and CX-3543 potentiate immunotherapies in CRC.

Another molecule interfering with rDNA transcription is BMH-21. BMH21 is a DNA intercalator molecule which impairs RNA pol I activity by binding to rDNA GC-rich regions and simultaneously disengages RNA pol I from rDNA chromatin and activates its proteasome-mediated degradation [188,189]. BMH-21 activates a rapid p53-dependent cytotoxic effect with little associated DNA damage in many human cancer cell lines, including HCT-116 CRC cells, and is also very potent in inhibiting HCT-116 xenograft growth in mice [190]. Very interestingly, it was recently reported that CRC patient-derived xenografts contain a sub-population of cancer stem cells characterized by high expression levels of the RNA pol I subunit A (PolR_1_A) and elevated biosynthetic capacities [191]. Besides, PolR_1_A is shown as one of the prerequisites for in vivo tumor growth [191]. CRC stem cells with high expression of PolR_1_A were classified at the top of tumor stem cell hierarchy and ip injection of BMH-21 induced a significant decrease in PolR_1_A-high stem cell and in tumor xenograft growth [191]. In addition, it appears that the FDA-approved antimalarial drug amodiaquine was recently shown to block rDNA transcription and proliferation of various human CRC cells lines by a mechanism very close to the induction of ribosome biogenesis stress and cell death by BMH-21 [192]. Similarly, the natural plant-derived product alkaloid haemanthamine was shown to specifically inhibit pre-rRNA processing, leading to the accumulation of the 47S pre-rRNA and impeding the formation of the mature 28S and 5.8S species [193]. Interestingly, haemanthamine was reported to trigger p53-associated nucleolar stress and apoptosis in CRC HCT-116 [193], arguing in favor of its use in CRC treatment. Collectively, these data indicate that inhibitors of ribosome biogenesis belonging to the CX-5436 family [194], BMH-21 and its derivatives [195], and plant alkaloids [193] hold great potential for CRC treatment and evidence of their efficacy in clinical phase I/II for human CRC treatment are eagerly awaited.

## 6. Conclusions

Evidence that the ribosome biogenesis pathway is altered in CRC has markedly increased in recent years. Studies using cellular and animal models largely contributed to establishing that the alterations leading to quantitative increase in ribosome production were linked to colorectal tumorigenesis initiation and/or progression. At present, clinical studies have also highlighted several regulators of ribosome biogenesis as novel biomarkers of human CRC, reinforcing the role of ribosome biogenesis in CRC. However, while few studies initiated the long process of demonstrating that the ribosome biogenesis pathway is an innovative target in CRC management, pre-clinical studies on animal models with various chemical or natural inhibitors of ribosome biogenesis are now needed to determine their objective effectiveness and to address their potential benefit(s) for CRC patients. Moreover, research is ongoing in many laboratories worldwide to unravel new ribosome biogenesis inhibitors following various drug discovery strategies, from drug-repurposing to development of high-throughput screening. These strategies may identify molecules to target not only the increase in ribosome biogenesis observed in CRC cells but also the cancer-modified ribosomes, the recent discovery of which is owing to the tremendous progress that has recently been made to obtain high-resolution structural ribosomal features. It is compelling to find out that several molecules that were initially discovered with potent anti-CRC effects (i.e., catalpol, calcimycin, flavonoid derivatives, oxaliplatin) are target ribosome biogenesis [196]. Thus, a new phase in CRC patient management is envisioned, where characterization of ribosome biogenesis pathways together with qualitative analysis of rRNAs, will help to create personalized anticancer molecules with much less genotoxicity.

## Figures and Tables

**Figure 1 cells-09-02361-f001:**
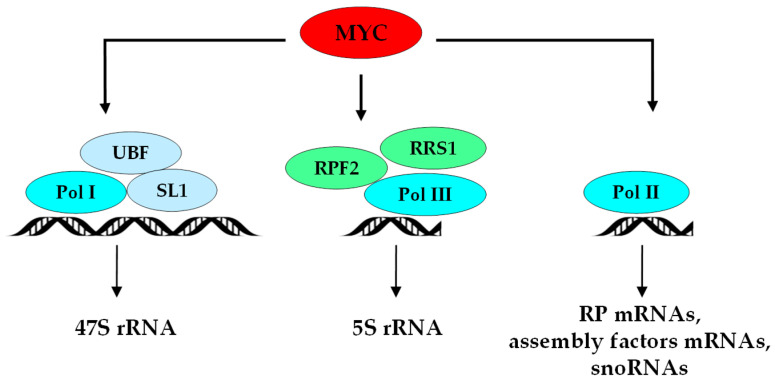
Regulation of early transcriptional steps of ribosome biogenesis in CRC. MYC is overexpressed in colorectal tumor cells, binds to rDNA sequences, and in cooperation with UBF and SL1 factors, leads to the hyperactivation of RNA pol I-mediated synthesis of the 47S rRNA precursor [84]. MYC, in cooperation with RPF2 and RRS1, activates RNA pol II-mediated transcription of genes coding for ribosomal assembly factors and RPs, as well as RNA pol III-mediated synthesis of 5S rRNA [57]. Overexpression of the three rRNA polymerases and the associated factors is frequent in CRC [80,81,82,83]. Together with RPF2 (protein ribosome production factor 2 homolog), RRS1 cooperates with POL III and drives 5S rRNA synthesis, while POL II activates ribosomal proteins and assembly factors mRNAs and snoRNAs synthesis.

**Figure 2 cells-09-02361-f002:**
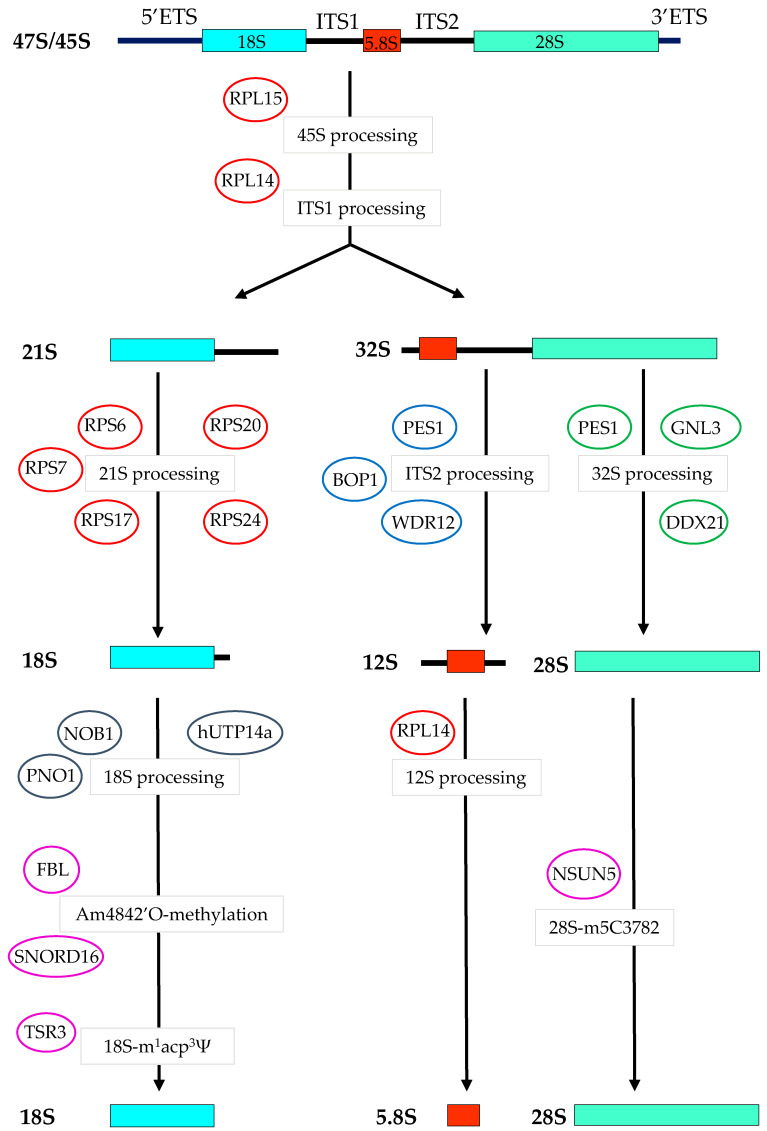
Schematic representation of ribosome biogenesis factors upregulated in colorectal cancer cells and their known sites of regulation of pre-rRNA processing pathways. The production of mature 18S, 5.8S, and 28S result from the synthesis and processing of a single 47S rRNA precursor characterized by noncoding 5′ and 3′ external transcribed spacers (ETS) and internal transcribed spacers (ITS1 and ITS2). In colorectal cancer cells, overexpression of the *45S* rRNA is a biomarker of poor prognosis [87]. RPL15 is involved in pre-rRNA processing at the internal transcribed spacer 1 (ITS1) site of the 47S pre-rRNA and is overexpressed in CRC [148]. RPL14 controls the processing of the 45S pre-rRNA and 12S rRNA and is highly expressed in CRC [140]. RPS6, RPS7, RPS 17, RPS20, and RPS24 are involved in the formation of the 18S rRNA and are overexpressed in CRC respectively, in References [132,139,141,142,150]. Pescadillo homolog 1 (PES1), block of proliferation (BOP1), and WD-repeat domain 12 protein (WDR12) are involved in the formation of the 12S rRNA and overexpressed in CRC respectively, in References [113,122,130]. PES1 is also involved with DDX21 and GNL3 in the processing of the 32S to the 28S rRNA and GNL3 is overexpressed in CRC [116]. RPL14, which is overexpressed in CRC [141], further activates the processing of the 12S rRNA to mature 5.8S. 18S pre-rRNA processing is activated by NIN1 (RPN12) binding protein 1 homolog (NOB1) in cooperation with “partner of NOB1” (PNO1) which is overexpressed in CRC [151]. 18S pre-rRNA processing is also activated by human U3 protein (UTP) 14a (hUTP14a), and is overexpressed and constitutes a marker of poor prognosis in CRC [152]. Base and nucleotide modifications are important modifications that control late steps of rRNA maturation. The human nucleolar enzyme NSUN5 catalyzes the C^5^ methylation of cytosine residue C_3782_ of the 28S rRNA (18S-m_5_C3782) and is upregulated in CRC and associated with disease progression [153]. The C/D-Box small nucleolar RNA 16 (SNORD16) guides fibrillarin (FBL) to methylate the 2′O-ribose on the 18S-Am484 site and constitutes a molecular marker of CRC and a driver of colorectal tumorigenesis [154]. The ribosome biogenesis protein TSR3 induces the 1-methyl-3α-amino-α-carboxyl-propyl pseudouridine (m^1^acp^3^Ψ) modification on uridine U1248 of the 18S rRNA and is overexpressed in CRC and associated with colorectal tumorigenesis [155].

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
