# Peer review of "Ribosome Biogenesis Alterations in Colorectal Cancer"

_cells, 2020, doi:10.3390/cells9112361_

Round 1

Reviewer 1 Report

This review combines diverse facts related to misregulated ribosome biogenesis in cancer cells to discuss the possibility of targeting this pathway in the clinical management of colorectal cancer. While the ribosome is certainly underappreciated as a regulator of gene expression in the cancer context, the review falls short of providing a clear rationale for targeting ribosome biogenesis as a promising approach in the CRC therapy. Part 5 ("Targeting ribosome RNA synthesis in colorectal cancer"), which would be expected to bridge basic research findings to the clinical side, appears more of a collection of random facts, often contradictory. For once, it is not clear why would targeting a housekeeping process such as Pol I transcription (if that is even the principal mechanism behind the CX compounds mentioned) be selectively toxic for cancer cells, wouldn't any other rapidly cycling cells (for example, the transit-amplifying cells of the intestinal epithelium) be the primary target as well? This section would benefit from some serious rewriting, with a clearly defined narrative and well argued conclusions.

Apart from the aforementioned deficiencies in the content, the review is written well. The authors should try to weed out occasional typos and grammar issues in the text, although this does not affect readability too much.

157 change to in addition to regulation?

167 the RP gene expression

193 contributes to

201-202 "mutations associated with dysfunctional ribosomes are associated with tissue-specific human pathologies" - awkward sentence construction

204 "these" alterations? I do not see how AgNOR staining is related to the preceding discourse on RPs

210 "could have become" not sure if this is the verb form the authors want to use here

233 maybe "in addition to contributing to"?

256 why "even" BOP1? Also, should be pre-rRNA cleavages, not rRNA.

274 drives

282 should be "promoted" or "promotes"?

360 please rearrange the sentence for better readability

439-443 "an excessive production of RP" This is misleading since the root cause in ribosomopathies is usually the reduced rather than excessive production of an RP. The same applies to nucleolar stress, the authors need to distinguish between the primary cause and the resulting excess of other RPs in the nucleus, which in turn may bind to extraribosomal targets such as MDM2.

479-480 "Therefore, it would be interesting to define the translational regulation associated with the increased synthesis of the 60S subunit in colorectal oncogenesis." I don't see how this follows from the decrease in 60S formation after knocking down RPL15 described immediately before this.

551-552. "Ribosome biogenesis processing factors that are upregulated in CRC are indicated on Figure 2" Do the authors imply that the factors they present in the diagram have any special role? Or is this just a random sample of factors that happened to be mentioned in the literature? Would the authors expect that most, if not all, ribosome biogenesis factors be upregulated in tumor cells as compared with the adjacent (mostly quiescent) cells in the normal tissue? In fact, earlier in the manuscript (lines 150-152), they note that " Hyperproliferative cancer cells are by definition cells with perturbed energy homeostasis and increased activity in protein synthesis [34-36,44], and thus, an increase in ribosome biogenesis participates in maintaining such high rate of protein synthesis [45][46][47][48]." Please clarify these issues in the text.

557 evidence. Also, please consider restructuring this sentence, the meaning of which is very hard to understand.

572 "on the four rRNAs" Is there evidence that 5S is methylated (which I assume is the fourth rRNA)?

616-645 The part about CX-5461 is extremely confusing and will require careful rewriting. The authors first call these drugs a breakthrough (line 625), only to say later that they remain untested in patients (line 645). They also claim that CX causes "stabilization of p53 and subsequent cell death" (line 637), and the next sentence seems to imply that this is not a good thing since the drug targets stem cells, while in line 640 they say apoptosis is induced in p53 mutant cells, which contradicts the mechanism dependent on normal p53 function as proposed earlier. Finally, in lines 655-656, the DDR is implicated as the mechanism of cell death, with ribosome biogenesis seemingly playing no role. All this makes things very confusing for the reader. What about this recent study: doi.org/10.1073/pnas.1921649117 ?

Author Response

October 18th, 2020 MS: “Ribosome biogenesis alterations in colorectal cancer” by Nait Slimane S et al. Reply to reviewer’s 1 analysis.

Dear reviewer,

Thank you for the careful assessment concerning our work and for the very useful comments. We have now rewritten part of the section “Targeting ribosome biogenesis in CRC” in order to make the role of CX-5641 clearer and less contradictory to the reader. The recent paper published by Bruno et al. on its implication as a topoisomerase II poison was mentioned in the original MS submitted, however it was not emphasized enough as a novel fact that implies reconsideration of its role. We hope we have now clarified this aspect.

As noted by the reviewer, most of the experiments with ribosome biogenesis inhibitors have been performed on CRC cell lines and clinical studies in CRC are indeed still lacking. However, the efficacy of some ribosome biogenesis inhibitors has been demonstrated in hematopoietic cancers and are tested in PhaseI/II clinical trials, and one can expect that it will lead to testing their effects in other types of cancer including CRC.

We have corrected the grammar errors and typos that have been highlighted by the reviewer and others that were not mentioned. Please find below the corrections (in bold) in the text according to reviewer1 remarks:

 157 change to in addition to regulation? Changed to “in addition to regulating”.

167 the RP gene expression. RPs corrected to “RP gene expression”.

193 contributes to : corrected accordingly.

201-202 "mutations associated with dysfunctional ribosomes are associated with tissue-specific human pathologies" - awkward sentence construction. Phrase changed to “ RP mutations producing dysfunctional ribosomes…”

204 "these" alterations? I do not see how AgNOR staining is related to the preceding discourse on RPs. “These” was suppressed.

210 "could have become" not sure if this is the verb form the authors want to use here: replaced with “could be a pathological…”

233 maybe "in addition to contributing to"? replaced as suggested.

256 why "even" BOP1? “even” was suppressed  Also, should be pre-rRNA cleavages, not rRNA; corrected as suggested to “pre-rRNA”.

274 drives: corrected.

282 should be "promoted" or "promotes"? replaced with “promotes”.

360 please rearrange the sentence for better readability: The sentence was changed to “ form with GNL3 a complex involved in late processing…..rRNA, before the incorporation of the latter into the 60S particle”.

439-443 "an excessive production of RP" This is misleading since the root cause in ribosomopathies is usually the reduced rather than excessive production of an RP. The same applies to nucleolar stress, the authors need to distinguish between the primary cause and the resulting excess of other RPs in the nucleus, which in turn may bind to extraribosomal targets such as MDM2. We have rephrased this paragraph according to the reviewer’s remark.

479-480 "Therefore, it would be interesting to define the translational regulation associated with the increased synthesis of the 60S subunit in colorectal oncogenesis." I don't see how this follows from the decrease in 60S formation after knocking down RPL15 described immediately before this. We have suppressed “therefore” and rephrased the sentence to clarify the relationship between RPL15 and the 60S production.

551-552. "Ribosome biogenesis processing factors that are upregulated in CRC are indicated on Figure 2" Do the authors imply that the factors they present in the diagram have any special role? Or is this just a random sample of factors that happened to be mentioned in the literature? Would the authors expect that most, if not all, ribosome biogenesis factors be upregulated in tumor cells as compared with the adjacent (mostly quiescent) cells in the normal tissue? In fact, earlier in the manuscript (lines 150-152), they note that " Hyperproliferative cancer cells are by definition cells with perturbed energy homeostasis and increased activity in protein synthesis [34-36,44], and thus, an increase in ribosome biogenesis participates in maintaining such high rate of protein synthesis [45][46][47][48]." Please clarify these issues in the text.

Most of ribosome biogenesis processing factors are indeed expected to be upregulated in cancers but only a few studies have described these regulations in CRC. Here we present the result of these CRC studies and we describe the known role of these processing factors in ribosome biogenesis. However and as mentioned in the text, ribosome biogenesis regulation was not systematically examined in these studies, and it does represent an important data that needs to be provided. Our aim is to indicate that these processing factors known to modulate ribosome biogenesis are interesting targets in potential CRC cell treatment.

557 evidence. Also, please consider restructuring this sentence, the meaning of which is very hard to understand. The sentence was reconstructed to “However, evidence that altered chemical modifications affect a targeted set of translated mRNAs during development and disease has provided further insight into the impact of qualitative rRNA modifications on ribosome function ».

572 "on the four rRNAs" Is there evidence that 5S is methylated (which I assume is the fourth rRNA)? The reviewer’s remark is right, 5S rRNA-ribose methylation has not been demonstrated, we corrected and deleted “four”.

616-645 The part about CX-5461 is extremely confusing and will require careful rewriting. The authors first call these drugs a breakthrough (line 625), only to say later that they remain untested in patients (line 645). They also claim that CX causes "stabilization of p53 and subsequent cell death" (line 637), and the next sentence seems to imply that this is not a good thing since the drug targets stem cells, while in line 640 they say apoptosis is induced in p53 mutant cells, which contradicts the mechanism dependent on normal p53 function as proposed earlier. Finally, in lines 655-656, the DDR is implicated as the mechanism of cell death, with ribosome biogenesis seemingly playing no role. All this makes things very confusing for the reader. The part about CX-5641 was rewritten to consider its role in different cellular context, i.e. p53-wild type vs. p53-mutant CRC cell lines.

What about this recent study: doi.org/10.1073/pnas.1921649117 ? This study was mentioned in the MS (ref 186) but is now given more importance regarding the reviewer’s remark.

We would like to thank the reviewer for its pertinent work and remarks concerning the MS and we hope we have addressed his remarks with relevance.

Sincerely yours,

Hichem C Mertani and Jean-Jacques Diaz

Reviewer 2 Report

This review covers thoughtfully the current knowledge on the role of ribosome biogenesis alteration in human colorectal cancer and explore the possibility of targeting ribosome production for therapeutic purpose. While other reviews can be found addressing the topic of ribosome biogenesis in cancer, this one has an advantage in that it focuses on colorectal cancer. No doubt it will be of interest to a wide audience.

Minor comments

1/ In addition to KRAS it might be worth mentioning BRAF mutation (BRAF V600E in particular) which belongs to the same signaling pathway and occurs in about 10% of patients with metastatic CRC. KRAS and BRAF mutations are generally mutually exclusive.

2/ In Figure 2, it might be worth indicating Myc targets and illustrating protein complexes as such.

Author Response

October 18th, 2020

MS: “Ribosome biogenesis alterations in colorectal cancer” by Nait Slimane S et al.

Reply to reviewer’s 2 analysis.

Dear reviewer,

Thank you for the careful assessment concerning our work and for the very useful comments.

  1. We have now mentioned in the main text the importance of BRAF mutations in metastatic CRC prognosis factor as recommended by the reviewer.

  1. For the figure 2 we have attempted to represent the Myc targets as suggested by the reviewer, however this led to a figure that would be interpreted as Myc being the prominent factor altered in ribosome biogenesis in CRC. We recognize the importance of Myc in the regulation of ribosome biogenesis (see figure1 and text) but for the figure2 we have decided to maintain the original presentation and hope the reviewer will understand our objective.

We would like to thank the reviewer for its pertinent work and remarks concerning the MS and we hope we have addressed his remarks with relevance.

 Sincerely yours,

 Hichem C Mertani and Jean-Jacques Diaz

Round 2

Reviewer 1 Report

line 442: Although the preceding part of the paragraph has been corrected, one inaccurate phrase remains: “Diseases associated with excessive RP production are known as ribosomopathies”. Please correct “excessive RP production”, this is misleading.

Author Response

Daer Editor,

 Thank you for your time and consideration, we were supposed to change the phrase before. It has been now corrected accordingly to "Diseases associated with RP gene mutations are known as ribosomopathies and are often linked to cancer predisposition".

 Sincerely yours,

 Hichem Mertani & Jean-Jacques Diaz